# Mental Health and Physical Complaints of German Children and Adolescents before and during the COVID-19 Pandemic: A Repeated Cross-Sectional Study

**DOI:** 10.3390/ijerph20054478

**Published:** 2023-03-02

**Authors:** Julia Hansen, Artur Galimov, Jennifer B. Unger, Steve Y. Sussman, Reiner Hanewinkel

**Affiliations:** 1Institute for Therapy and Health Research, IFT-Nord, Harmsstraße 2, 24114 Kiel, Germany; 2Tobacco Center of Regulatory Science, Department of Population and Public Health Sciences, Institute for Health Promotion and Disease Prevention Research, Keck School of Medicine, University of Southern California, 1845 N. Soto Street, Los Angeles, CA 90033, USA

**Keywords:** emotional problems, hyperactivity-inattention, conduct problems, internalizing problem behavior, externalizing problem behavior

## Abstract

The potential impact of the COVID-19 pandemic on young people’s mental and physical health is of increasing concern. We examined the levels of internalizing and externalizing problem behavior and physical complaints before and during the COVID-19 pandemic in Germany. Data came from a repeated cross-sectional study on child and youth health in schools in Germany. Assessments took place from November to February each year. Two data collections were conducted before the COVID-19 pandemic in 2018–2019 and in 2019–2020. Collections during the pandemic took place in 2020–2021 and 2021–2022. A total of 63,249 data observations were included in the analyses. Multilevel analyses were used to examine temporal trends in mean emotional problems (e.g., often unhappy, downhearted), hyperactivity-inattention (e.g., constantly fidgeting or squirming), conduct problems (e.g., fights with other children), and physical complaints. Models were adjusted for age, gender, school type, socioeconomic status, and sensation seeking. During the COVID-19 pandemic, children and adolescents in Germany experienced an increase in emotional problems from the pre-pandemic cohort 2019–2020 to the pandemic cohort 2021–2022 (β = 0.56, 95% CI (0.51–0.62)) and, over the course of the pandemic, reported elevated levels of physical complaints (β = 0.19, 95% CI (0.16–0.21)). Findings of increased emotional problems and physical complaints after the two years of the pandemic support the ongoing demand for low-threshold health promotion and prevention and the need for further monitoring of young people’s health in Germany.

## 1. Introduction

Studies examining the impact of the pandemic on the population’s mental health status observed an increase in internalizing problems, such as depressive and anxiety symptoms, as well as a worsening trend in general mental health [1,2,3]. In Spring 2022, two years after the outbreak of the COVID-19 pandemic, the World Health Organization (WHO) stated that the pandemic triggered a 25% increase in the prevalence of internalizing problems, such as anxiety and depression worldwide, possibly due to the unprecedented stress caused by social isolation and fear of infection resulting from the pandemic [4]. Systematic reviews on the impact of COVID-19 demonstrate that children and adolescents across the world face mental health deterioration due to both the onset of the COVID-19 pandemic and related control measures [5,6,7]. Despite being less studied, emerging evidence also indicates an increase in somatic health complaints [8].

Children and adolescents with low socioeconomic status (i.e., children from families with low education levels or limited financial resources) are at higher risk of suffering from COVID-19 lockdown consequences, which is in line with research on social inequality and mental health [9]. In contrast to males, females showed higher depression and anxiety levels during the COVID-19 pandemic [10]. Evidence suggests that the prolonged pandemic situation and the related societal restrictions have had an impact on females in particular and that the rise in internalizing problem behavior and health complaints can, in part, be attributed to an increase in loneliness [11].

Studies from Germany indicate that children and adolescents who felt highly burdened in the initial phase of the pandemic in 2020 had a significantly higher risk of mental health problems such as internalizing problems (e.g., emotional problems and depression symptoms) and externalizing problems (e.g., hyperactivity-inattention symptoms and conduct problem behavior), suffered from physical complaints (e.g., headache and back pain), and had a lower quality of life, compared with pre-pandemic data [9,12,13,14,15,16]. Furthermore, a distinct weight gain during COVID-19 was found in German children and adolescents, which may lead to physical complaints [17].

The above-mentioned results indicate an increase in internalizing and externalizing problem behavior and physical complaints in children and adolescents during the pandemic. However, there is still a need for research on the vicissitudes of health status, as it is not yet clear whether the consequences of the pandemic should be classified as acute or long-term and can lead to enduring disorders over time. According to a Dutch five-wave prospective cohort study, adolescents reported more internalizing symptoms during the lockdown in February 2021 than in previous pandemic phases [18]. Other studies reported that internalizing and externalizing problem behavior in children and adolescents increased during strict measures (e.g., COVID-19 lockdown) [19]. There is limited evidence on the consequences of the prolonged pandemic situation on German children’s and adolescents’ internalizing problem behavior, externalizing problem behavior, and physical comfort. In 2023, in a recent rapid review of changes in the mental health of children and adolescents in Germany, the German federal government agency and research institute responsible for disease control and prevention (RKI) called for close and continuous monitoring of children’s mental health and improved identification of risk groups during the COVID-19 pandemic. A need for action was identified since a significant increase in mental distress and symptoms during the pandemic was observed, knowing that the prevalence of mental health problems in children and adolescents had declined in times before the pandemic. The RKI also identified a lack of studies monitoring the mental health of youths in Germany during the pandemic and beyond [20]. Therefore, we wanted to add to the existing body of knowledge and refine existing results with findings from recently collected data. We aimed to determine (1) whether internalizing problem behavior (emotional problems), externalizing problem behavior (hyperactivity-inattention and conduct problems), and somatic health complaints (headache, back pain, and stomach ache) increased from before the pandemic to year 1 and also to year 2 of the pandemic, and (2) whether certain groups (e.g., girls) were disproportionately affected by the pandemic, as evidenced by a greater increase in symptoms. Specifically, we hypothesized that children’s and adolescents’ symptoms/levels of emotional problems, hyperactivity-inattention, and conduct problems increased during the COVID-19 pandemic compared to pre-pandemic trends. We assumed that elevated levels would also be evident beyond the initial phase of the pandemic: We expected an increase in the first year of the pandemic because the COVID-19 control measures were significantly stricter than during the further course of the pandemic, and, with the normalization of everyday life, no further increase in the elevated levels. Conversely, we assumed that children and young people reported physical complaints (e.g., back pain) more frequently during the course of the pandemic since factors relevant to the complaints, such as weight gain, only become apparent after a certain period of time.

## 2. Materials and Methods

### 2.1. Study Design & Setting

The German “Präventionsradar”, an ongoing school-based study, was introduced in the school year 2016/2017 to monitor children and adolescent health behaviors in Germany on an annual basis [21]. Invitations to participate in the study were emailed to all German schools in 13 federal states that approved the study (convenience sampling). Ethical approval was obtained from the German Psychological Society (RH 042015_1).

Any secondary school located in the 13 federal states had the opportunity to register classes from the 5th to the 10th grade for participation. Schools were encouraged to participate every year. Every class member of a registered class was eligible to participate in the study. Participation was voluntary. Informed written consent from parents and students’ verbal assent were collected prior to administering the survey. All participants had the option of withdrawing from the study at any time without penalty.

### 2.2. Data Collection

The data collection via web-based questionnaire took place every school year from November to February. Pseudonymized data were collected in a classroom setting by trained research staff or instructed school personnel through self-completed web-based questionnaires during one classroom hour (45 min). The web-based questionnaire (in-house development) was accessible via ift1.de, a proprietary software solution. The students accessed the website ift1.de and activated the questionnaire with a six-digit web code that was randomly assigned by the teacher. Multiple entries from the same individual were prevented during the data collection process by using the web codes. Throughout the COVID-19 lockdown, students were instructed to complete the survey at home during online classes. The teacher distributed the link to the questionnaire and web codes during the digital class. Parents were instructed not to look at their children’s responses. Participating schools received a report on the health behavior of their student body to improve their prevention programs.

### 2.3. Study Population

The repeated cross-sectional study was undertaken using four waves (2018/2019–2021/2022) of “Präventionsradar” data. The figure below shows the data collection time points, the N of each data collection, and the related pandemic phases in Germany (see Figure 1).

The majority of children and adolescents (N = 33,601) participated only in one wave. Participation in two of the four waves was N = 9732; in three of four waves was N = 2400, and in all four waves was N = 746. Further, 10th graders were not followed up again. In total, the 63,249 datasets came from 46,479 children and adolescents.

Response rates were as follows for the four survey waves: In wave 3 (2018–2019), 14,875 young people viewed the questionnaire, the participation rate was 95.7%, and the completion rate was 91.6%. In wave 4 (2019–2020), 18,392 young people viewed the questionnaire, the participation rate was 91.6%, and the completion rate was 89.4%. In wave 5 (2020–2021), 15,652 young people viewed the questionnaire, the participation rate was 91.3%, and the completion rate was 88.3%. In wave 6 (2021–2022), 19,413 young people viewed the questionnaire; the participation rate was 92.1%, and the completion rate was 83.9%. The reasons for drop-out were primarily technical problems in the schools (e.g., unstable internet connection, device-based time-out settings) and the lack of time to complete the questionnaire during the lessons.

### 2.4. Variables and Measures

Emotional problems, hyperactivity-inattention, conduct problems, and physical complaints: The Strengths and Difficulties Questionnaire (SDQ) is the most widely used mental health screening instrument for children and adolescents, assessing internalizing and externalizing problem behaviors [23]. We used a validated German version of the SDQ [24]. Emotional problems representing internalizing behavior were assessed using the 5-item emotional problems subscale of the SDQ (e.g., “I am often unhappy, depressed or tearful”). Hyperactivity-inattention representing one dimension of externalizing behavior was assessed using the 5-item hyperactivity-inattention- subscale of the SDQ (“I am restless, I cannot stay still for long”). Conduct problem behavior, another dimension of externalizing behavior, was assessed using the 5-item conduct problems subscale (e.g., “I take things that are not mine from home, school, or elsewhere”). Response options were on a 3-point answer scale ranging from “not true” (0), “somewhat true” (1), and “clearly true” (2). Scores of the SDQ subscales range between 0 and 10. Higher scores represent more severe problem behavior/symptoms.

Physical complaints were assessed by the frequency of experiencing headaches, back pain, and stomach ache. Response options were “seldom or never” (1), “once a month (2)”, “once a week” (3), “several times a week” (4), and “daily” (5) [25]. Variables were summed up, and a mean score was calculated for each observation. The scale ranges from 1 to 5. Higher scale scores suggest more frequent physical complaints. Emotional problems, hyperactivity-inattention, conduct problems, and physical complaints/symptoms are the primary outcomes/endpoints of this study. (Cronbach’s Alpha of the scales between 0.7–0.8).

Sociodemographic variables: Self-reported sociodemographic variables were age, gender, and subjective social status (SSS) measured with the MacArthur Scale. (“Please place an ‘X’ on the rung that best represents where you think your family stands on the ladder?” (on a 10-point scale corresponding to a picture of a ladder, ranging from zero (low income, the worst jobs, the lowest education) to 10 (high income, the best jobs, the highest education)) [26]. Age was a continuous variable, and gender was coded boys = 0, girls = 1. For descriptive analyses, the SSS was split at the median. Sociodemographic variables also include school type (not self-reported). Gymnasium is the most advanced type of secondary school that strongly emphasizes academic learning. School type was recoded in others = 0, gymnasium = 1.

Sensation seeking: Sensation Seeking was assessed to control for a stable personality trait. It was measured with two items: “How often do you do dangerous things to have fun?” and “How often do you do exciting things, even if they are dangerous?” (each on a five-point scale ranging from “not at all” (1) to “very often” (5)) [27]. An index of sensation seeking was constructed by averaging responses to these two items. For descriptive analyses, sensation seeking was split at the median into low and high sensation seeking traits.

### 2.5. Statistical Analysis

First, mean SDQ scores were calculated. Descriptive data were weighted to census data to compensate for bias. Hence, a weighting factor was calculated to address selection mechanisms during recruitment. The weighting factor was created based on data from the Federal Statistical Office and takes age, gender, and type of school into account.

Second, repeated cross-sectional analyses were used to examine temporal trends in mean emotional problems, hyperactivity-inattention and conduct problems, and physical complaints. Four multilevel mixed-effect linear regression models were fitted to analyze the changes in the outcomes over time (Level 1), accounting for random intercepts at the school level (Level 3) and within individuals (Level 2). Unadjusted models were set up first. Next, adjusted models were set up to test the main effects. All four models were adjusted for sociodemographic variables and sensation seeking.

Third, three-way interactions (time × age × gender) were analyzed. For the interaction terms, age was recoded into two categories, including children aged from 9 to 12 years and adolescents aged from 13 to 18 years. Bonferroni adjusted pairwise comparisons of predictive margins were used to indicate differences between groups and time points. The regression curves were presented graphically.

Children and adolescents with missing data on the primary outcomes were not analyzed (see Appendix A). To test the robustness of findings, repeated analyses without random intercepts (Level 2) were carried out. All statistical analyses were conducted using Stata software (version 17.0; Stata Corp, College Station, TX, USA). Beta coefficients (βs) with 95% CIs were reported with statistical significance set at *p* < 0.05 (2-tailed).

## 3. Results

### 3.1. Participants

The total N at each wave was: 2018–2019: N = 14,242; 2019–2020: N = 16,843; 2020–2021: N = 14,287; 2021–2022: N = 17,877. In sum, N = 63,249 data observations were collected over the four years (for further details, see Appendix A). Table 1 shows the profiles (unweighted) of each cross-sectional sample. The mean age at each wave was 13 years (range 9–18). An equal gender distribution was achieved each year.

### 3.2. Descriptive Data

The mean emotional problem subscale score increased over time, from 2.89 (95% CI (2.84–2.93)) in 2018–2019 and 2.96 (95% CI ((2.92–3.00)) in 2019–2020 to 3.29 (95% CI (3.25–3.34)) in the 2020–2021 cohort and 3.48 (95% CI (3.44–3.52)) in 2021–2022 (see Appendix A).

The mean hyperactivity-inattention score was 3.55 (95% CI (3.51–3.58)) in 2018–2019, 3.64 (95% CI (3.61–3.68)) in 2019–2020, and 3.74 (95% CI (3.70–3.77)) in the 2020–2021 cohort, and 3.85 (95% CI (3.82–3.88)) in the 2021–2022 cohort (see Appendix A).

The mean conduct problem score before the pandemic was 2.06 (95% CI (2.03–2.09)) in 2018–2019 and 2.19 (95%CI (2.16–2.22)) in 2019–2020. During the first year of the pandemic (2020–2021), the mean score was 2.14 (95% CI (2.11–2.17)), and in the second year (2021–2022), 2.13 (95% CI (2.11–2.16), see Appendix A).

Physical complaints mean score was 1.76 (95% CI (1.75–1.78)) in 2018–2019, 1.76 (95% CI (1.75–1.7)) in 2019–2020, 1.76 (95% CI (1.74–1.77)) in 2020–2021, and 1.93 (95% CI (1.91–1.94)) in 2021–2022 (see Appendix A). All reported mean scores were weighted to census data to compensate for bias.

### 3.3. Internalizing and Externalizing Problem Behavior from 2018–2019 to 2021–2022

Emotional problems: In the unadjusted multilevel model, a significant main effect of time was observed (β = 0.29, 95% CI (0.27–0.31)). Emotional problems increased over time from one wave to another. After adjusting for age, gender, SSS, school type, and sensation seeking, there are significant differences between the waves (β = 0.21, 95% CI (0.20–0.23)), but no significant main effect of time in the pre-pandemic time period from 2018–2019 to 2019–2020 was found. However, emotional problems increased significantly from the pre-pandemic cohort 2019–2020 to the pandemic cohort 2020–2021 (β = 0.39, 95% CI (0.34–0.44)) and to the 2021–2022 cohort (β = 0.56, 95% CI (0.51–0.62)); see Figure 2). Girls’ emotional problems in the pandemic period were significantly higher in 2021–2022 compared to 2020–2021 among both children and adolescents. This effect was not found in male participants. Significantly elevated levels in year 2 (2021–2022) were still apparent for all groups except for boys aged 9–12.

Hyperactivity-inattention: A significant but slight increase over the years was found in the unadjusted multilevel model (β = 0.10, 95% CI (0.08–0.11)) and in the adjusted model (β = 0.09, 95% CI (0.07–0.11)). The greatest increase was observed at the beginning of the pandemic from 2019–2020 to 2020–2021. Significantly elevated levels in year 2 compared to pre-pandemic levels were found for both 9–12-year-old and 13–18-year-old girls and 13–18-year-old boys but not for 9–12-year-old boys (see Figure 3).

Conduct problems: A significant main effect of time was observed (β = 0.02, 95% CI (0.01–0.04)) in the unadjusted multilevel model. After covariate adjustment, conduct problems in 2021–2022 were similar to the pre-pandemic level in 2018–2019. Overall, no elevated levels in the pandemic phases compared to pre-pandemic phases were observed (see Figure 4).

### 3.4. Physical Complaints from 2018–2019 to 2021–2022

In the unadjusted model, a significant main effect of time was observed (β = 0.09, 95% CI (0.08–0.10)), indicating that physical complaints remained at the same extent in the pre-pandemic period (1.72 95% CI (1.70–1.75), 1.74 95% CI (1.71–1.76)) and rose to 1.77 (95% CI (1.74–1.79)), and 1.99 (95% CI (1.96–2.01)) in the pandemic period. This effect remained significant in the adjusted model (β = 0.19, 95% CI (0.16–0.21)). In 2021–2022 a higher proportion of children and adolescents reported more frequent physical complaints compared to the previous three time points. Girls experienced more frequent physical complaints compared to boys. The greatest increase in physical complaints was observed in 13–18-year-old girls (contrast 0.39, 95% CI (0.33–0.43), see Figure 5).

### 3.5. Sensitivity Analyses

Repeated analyses without random intercepts at Level 2 revealed results in the same direction as mentioned above.

## 4. Discussion

By using 63,249 data observations from before and two years into the COVID-19 pandemic, this study provides insight into the levels of emotional problems, hyperactivity-inattention, conduct problems, and physical complaints of children and adolescents during the pandemic in Germany and, therefore, the potential effects of the pandemic on children’s and adolescents’ lives. Emotional problems slightly increased compared with pre-COVID-19 trends. Hyperactivity-inattention symptoms were still elevated in year 2 of the pandemic. Conduct problem behavior was not elevated in the pandemic phases. The physical complaints elevated during the pandemic, which indicates a more frequent experience of headaches, back pain, or stomach ache in year 2 of the pandemic. Overall, the study results support previous findings that indicate impairment of mental health, especially emotional problems (e.g., feeling unhappy, depressed, downhearted) for young people in Germany [9,13,14,16] and refine the existing findings to include results from the recent stages of the pandemic in a large-scale, school-based sample in Germany.

We found evidence that children and adolescents experienced more frequent symptoms of emotional problems during the pandemic. They felt more often depressed, unhappy, downhearted, anxious, and had worries. This is in line with existing research examining the internalizing behavior throughout the COVID-19 pandemic [1,2,10,28,29,30,31]. The results may be explained by the required large-scale behavioral changes from children and adolescents due to infection control measures which have been vital to tackling this pandemic [32], and the adverse effects of social isolation and disruptions in mental health care services due to the pandemic [33,34]. In addition, adolescence is an important stage of identity development, and young people are vulnerable to the influence of social environments, which in time of infection control measures, provided fewer opportunities and protective factors and more potential threats. Moreover, changes in daily routine may be particularly harmful to children and adolescents [35].

Internalizing problem behavior among the same age group increased over the years. Elevated levels were still above pre-pandemic levels in the second year of the pandemic, suggesting that the increase cannot be interpreted as a short-term increase, except in 9–12-year-old boys. We did not find elevated values for this group in the 2nd year. Therefore, the results do not fully support our hypothesis that no further increase in elevated levels will be observed as everyday life returns to normal. For the group of 9–12-year-old boys, the assumption seems to be correct. Females aged 13–18 had the highest average score of emotional problems, suggesting a shift from pre-pandemic normal scores to borderline/abnormal scores during the pandemic. With regard to gender-specific differences, girls perceived the pandemic as having a more drastic and global impact on their lives than boys, so the negative effects of the pandemic were more pronounced in girls [36]. This could be an explanation for the different developments of the groups.

We found a slight increase in hyperactivity-inattention symptoms for both male and female adolescents at the beginning of the pandemic from 2019–2020 to 2020–2021, which was also found by Daniunaite et al. [37]. In year 2 of the pandemic, hyperactivity-inattention symptoms were still more often reported across age groups, but findings indicate no further increase in pandemic year 2 compared to pandemic year 1. Our results, therefore, promise that inattention/hyperactivity symptoms could decrease with normalization of life, as symptoms were exacerbated during periods of lockdown, such as school closures, when children and adolescents were at home with distracting surroundings (e.g., TVs, phones, family members). Different from previous findings [29], we did not find elevated levels of conduct problems in general that can be attributed to the pandemic since levels are not significantly different from pre-COVID-19 levels. However, in the case of conduct problems, it is of particular relevance which infection control measures were imposed at the time of data collection. In times of homeschooling and few social contacts, conduct problems are no longer reliably identified with statements such as “I fight a lot.” In this regard, the results are inconsistent, and further monitoring is urgently needed to gain a better understanding of pandemics’ potential impact on conduct problems in youth.

Elevated levels of physical complaints became more apparent in the second year of the pandemic. Children and adolescents more often reported having headaches, back pain, and stomach ache. Specifically, for females aged 13–18, this means that the symptoms occurred with an average frequency of about once a month before the pandemic but tended to shift towards once a week during the pandemic. On the one hand, more frequent symptoms of physical complaints such as headache can be a consequence of a COVID-19 infection and raise concerns regarding chronic COVID symptoms [38]. On the other hand, the lack of exercise as an aftermath of lockdown measures [39,40] may lead to more frequent back pain among children and adolescents, as has already been shown for adults [41]. Some healthcare professionals raised concerns that certain reported symptoms might be a consequence of rigid social restrictions rather than direct consequences of COVID-19 infection [38,42]. Physical complaints in children, especially younger ones, can also be an expression of psychological stress, which increased for young people in Germany during the pandemic compared to the pre-pandemic time [9].

### Limitations of the Study

We would like to point out the following limitations: The results are self-reported and may have been influenced by response biases such as non-response bias and social desirability. It is possible that only German-speaking children and adolescents who attend schools that value school health promotion and prevention work were eligible to participate. As can be seen in Figure 1, the data were collected during the pandemic phases with strict measures, including school closures. This means that data from the surveys in the pandemic years were collected more frequently than in previous years as part of digital teaching. Although the parents were instructed not to be present when the data were collected, this confounding factor, which could have led to underreporting, cannot be ruled out. In addition, Germany may have had other infection control measures and COVID-19 stages at different time points than other countries. Thus, findings may not be generalizable to countries other than Germany or other samples. Differences in mental health before and over the course of the pandemic are attributed to the pandemic and interpreted as an effect of it. However, due to the repeated cross-sectional design, no causality can be derived. Several other individual and societal factors, as well as different analysis samples, may have influenced these differences. It is also possible that the Ukrainian–Russian war that began on 24 February 2022 had an impact on the mental health of young people. However, it should be noted that this could only be true for a very small proportion of the 21/22 sample, as most of the data were collected before the start of the war. Concerning the changes in the analytic sample over time, potential differences were countered by calculating a weighting factor based on the Federal Statistical Office, which at least reduces systematic selection bias. Furthermore, a number of variables were controlled for in the models, which strengthened the study results’ validity. The SDQ is a brief, short, and easy-to-use emotional and behavioral screening questionnaire for children and adolescents that can globally assess behavioral problems and behavioral strengths. However, we only used three of the instrument’s five subscales, which raises the question of validity, particularly for internalizing behaviors. Hence, we interpret the findings only as an indicator of change in one dimension of internalizing behavior (that of emotional problems). Clinical diagnoses of mental disorders (e.g., depression) do not result from this screening instrument.

## 5. Conclusions

Overall, the findings of this large-scale, school-based, repeated cross-sectional study confirmed a deterioration in mental health during the COVID-19 pandemic among young people in Germany, which was particularly evident in higher levels of emotional problems (e.g., often unhappy, depressed, downhearted). Since higher levels were also evident during the course of the pandemic, the increase cannot be interpreted as short-term for most young people. However, it is not yet clear how many young people have developed clinically relevant depression. The findings also refine existing evidence by indicating that physical complaints (e.g., headache, back pain) become more apparent as the pandemic progresses than in the early days of the pandemic. The Robert Koch Institute (RKI), one of the most important bodies for the safeguarding of public health in Germany, claims that routinely conducted trend and cohort studies would be desirable to survey the mental health of young people during the course of the pandemic and beyond [20]. Therefore, the study findings are highly relevant to public health and policy in Germany as implementation of infection control measures in Germany already takes children’s mental health into account. Taken together, the findings also highlight the need for low-threshold health promotion and prevention, especially for young people particularly affected by the pandemic.

## Figures and Tables

**Figure 1 ijerph-20-04478-f001:**
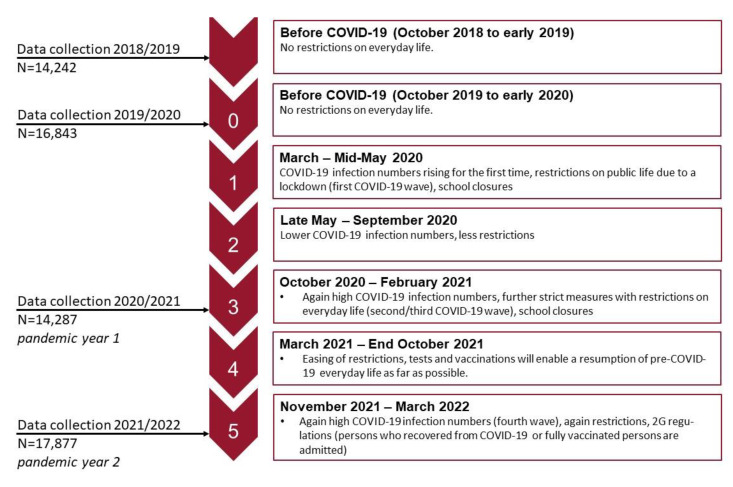
Pandemic stages [22] and data collection time points.

**Figure 2 ijerph-20-04478-f002:**
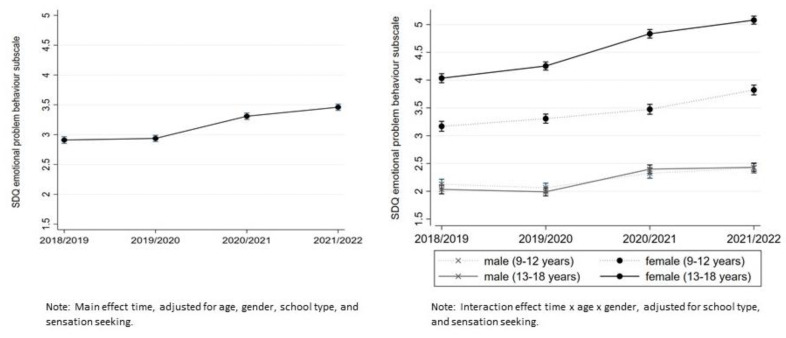
Mean SDQ subscale emotional problems scores from 2018–2019 to 2021–2022, cross-sectional measures, 2018–2019 and 2019–2020 pre-pandemic, 2020–2021 and 2021–2022 pandemic. The scale ranges from 0 to 5 (abnormal problem behavior).

**Figure 3 ijerph-20-04478-f003:**
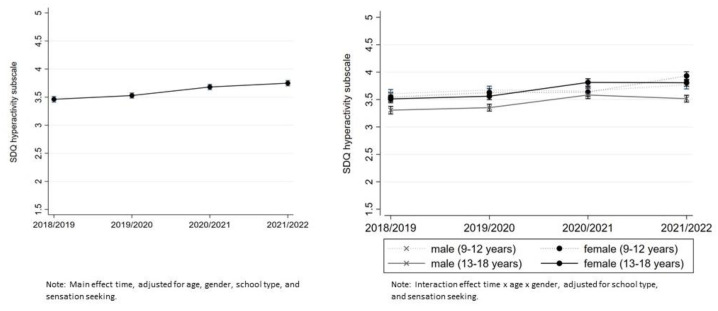
Mean SDQ subscale hyperactivity-inattention scores from 2018–2019 to 2021–2022, cross-sectional measures, 2018–2019 and 2019–2020 pre-pandemic, 2020–2021 and 2021–2022 pandemic. The scale ranges from 0 to 10 (abnormal problem behavior).

**Figure 4 ijerph-20-04478-f004:**
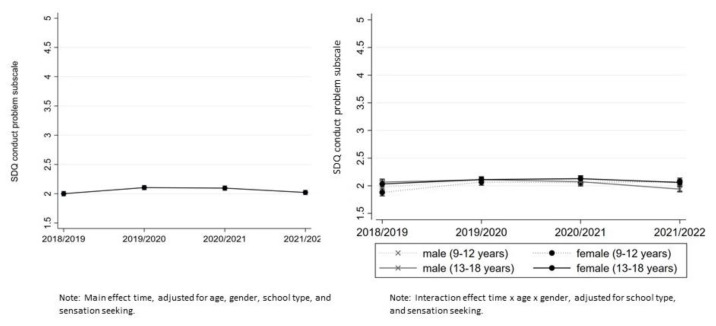
Mean SDQ subscale conduct problem scores from 2018–2019 to 2021–2022, cross-sectional measures, 2018–2019 and 2019–2020 pre-pandemic, 2020–2021, and 2021–2022 pandemic. The scale ranges from 0 to 10 (abnormal problem behavior).

**Figure 5 ijerph-20-04478-f005:**
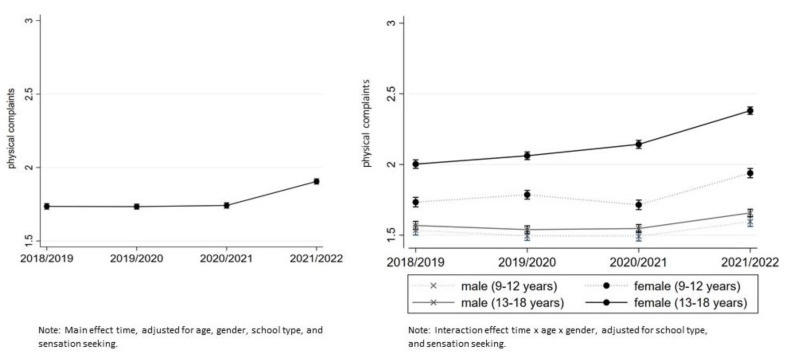
Physical complaints scores from 2018–2019 to 2021–2022, cross-sectional measures, 2018–2019 and 2019–2020 pre-pandemic, 2020–2021 and 2021–2022 pandemic. The scale ranges from 1 (never) to 5 (daily reported physical complaints).

**Table 1 ijerph-20-04478-t001:** Profile of each cross-sectional analysis sample from 2018–2019 to 2021–2022 of “Präventionsradar”, by year.

	Before the Pandemic		During the Pandemic	
	2018–2019	2019–2020	2020–2021	2021–2022
	N (%)			
Total sample	14,242 (100)	16,843 (100)	14,287 (100)	17,877 (100)
Gender				
Girl	6931 (48.9)	8195 (49.4)	7044 (49.9)	8685 (49.7)
Boy	7253 (51.1)	8379 (50.6)	7074 (50.1)	8775 (50.3)
School type				
Gymnasium	6386 (44.8)	9952 (59.1)	9107 (63.7)	10,771 (60.3)
Other	7856 (55.2)	6891 (40.9)	5180 (36.3)	7106 (39.7)
	**M (SD)**			
Age in years	13.0 (1.79)	13.1 (1.79)	13.0 (1.76)	13.1 (1.74)
SSS	6.8 (1.53)	6.8 (1.53)	7.0 (1.52)	7.0 (1.52)
(Scale 1–10)				
Sensation Seeking	2.3 (1.10)	2.3 (1.10)	2.3 (1.08)	2.4 (1.11)
(Scale 1–5)				

Sample sizes, gender, age, and school type are true (unweighted). N = Total Number of Observations; M = Mean; % = Percentage; SD = Standard Deviation; SSS = Subjective Social Status.

## Data Availability

The data that support the findings of this study are available from the corresponding author (J.H.) upon reasonable request.

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
