# Peer review of "Mental Health and Physical Complaints of German Children and Adolescents before and during the COVID-19 Pandemic: A Repeated Cross-Sectional Study"

_ijerph, 2023, doi:10.3390/ijerph20054478_

Round 1

Reviewer 1 Report

In the abstract, it is not clear, what was meant by "emotional problems". I think, it should be written in detail, what exactly was found by the means of what was measured.

In the abstract, the last sentence should be a conclusion but it sounds like a discussion. I would suggest to summarize your results in conclusions and move the discussion to the "Discussion" section.

If sentences on the lines 77 - 87 are dedicated for hypothesis, I would suggest to write them in the past tense because, I guess, after the analysis of your results you should already know if your hypothesis are true or false.

In my opinion, words 'Corona' should be changed to COVID-19 in figure 1.

I think, the description of the tests used for statistical analysis should be provided in the 'Methods' section. Currently, it is not clear how distribution of participants of this study was assessed.

In the 'Results' section the reference to the appendix is mentioned but I cannot find the appendix.

A mix of N with means and percentages with standard deviations in the table 1 may confuse the readers. I would suggest to split the table into two subsections and do not mix the measures. In addition, all of the abbreviations need to be explained.

I think, at the end of discussion the "Limitations of the study" subsection should be added.

Conclusions should summarize your results. Now, you talk about things you did not study in "Conclusions".

Reviewer 2 Report

Abstract: please remove [] from abstract because it is used only for references in MDPI journals.

Also include zeros before values, for example, .19 should be 0.19.

Introduction: Although the introduction part is well-structured, I urged the authors to include some debate on how this study enriches the existing body of knowledge and why this study in the German context is important.

Discussion: The limitation pat should appear after the discussion. Moreover, it should be a stand-alone section and should not be merged with the discussion of the results.

Similarly, it is important to highlight how your study is distinct from the existing literature and why it relates (or not relates) to already published material.

Additionally, the discussion section should highlight the important theoretical and practical implications.

Conclusion: this part needs more attention because it is written by including very less information. I personally feel that this section needs more attention from the authors and requires reconsideration.

Overall note: I am of the opinion that the study has been conducted in a decent manner. It is well-structured and well-written and requires some polishing in a few areas. I recommend its publication after minor revision. Best of Luck to the authors. 

Reviewer 3 Report

Dear authors,

I appreciate your effort in collecting the data from a large sample and following up periodically. The introduction was well-written. 

I want to comment on some points that would be better explained. 

First, I could not understand that these data were from the same sample or that these children were different each year. I recommend you add the participants to explain the characteristics and inclusion and exclusion criteria. 

There are so many measurement details that should be shortened and given the reliability results. 

You also discussed some other limitations; some other individuals and social factors could influence these results. In 2022, the Ukrainian-Russian war occurred, which might also affect the children and adolescents in Germany. I recommend adding this point to the discussion part. 

I wish you success in your work. 

Reviewer 4 Report

This manuscript highlights the impact of COVID-19 pandemic on the physical as well as mental health of German children and adolescents. Authors compared the mental and physical health data of children 9-18 years before and during the pandemic. They observed an increase in the emotional problems, hyperactivity-inattention and physical problems during the pandemic (2020-2021 and 2021-2022) compared to pre-pandemic phases (2018-2019 and 2019-2020). The manuscript is well-written for the most part and adheres to the journal’s guidelines. Strengths of the study includes (1) a large sample size, (2) adequate statistical analysis and (3) importance of the research area. Study limitations are adequately described in the discussion section of the manuscript.

Major issue:

I was not able to find the supplementary files related to this manuscript (Figure S1, Table S1, Table S2, Table S3, and Table S4). Therefore, authors are advised to upload the supplementary material associated with this submission.

Minor revisions:

1.      Line 18, delete “N=” from the sentence

2.      It would be better to add a leading zero before the decimal point

3.      It would be better to divide first section (“Study Design and Setting”) of methods into 3 or 4 subsections e.g. study design and settings, study population, Sampling, ethical considerations etc.

4.      Indicate what sampling technique was employed to recruit study participants in this repeated cross-sectional study.

5.      Indicate the platform/software used to develop and disseminate the e-questionnaire.

6.      How the multiple entries from the same individual were prevented during the data collection process?

7.      Kindly indicate that you have used a validated German version of SDQ in your study.

8.      How many children or their family members were infected with SARS-CoV-2 during the pandemic?

9.      Line 240; Delete “Main results:” from the title of this subheading.

Round 2

Reviewer 4 Report

Thanks to the authors for submitting the revised manuscript. The revision answers gaps identified in the first review, however, one key issue still remains and needs to be addressed before the manuscript is publishable

1. As this was an online survey, I would highly recommend you to provide the response rates of this survey (view rate, participation rate, completion rate etc.).

Author Response

As this was an online survey, I would highly recommend you to provide the response rates of this survey (view rate, participation rate, completion rate etc.).

We provided information about the response rates and added the following paragraph to the methods section:

“Response rates were as follows for the four survey waves: In wave 3 (2018-2019), 14,875 young people viewed the questionnaire, the participation rate was 95.7% and the completion rate was 91.6%. In wave 4 (2019-2020), 18,392 young people viewed the questionnaire, the participation rate was 91.6% and the completion rate was 89.4%. In wave 5 (2020-2021), 15,652 young people viewed the questionnaire, the participation rate was 91.3% and the completion rate was 88.3%. In wave 6 (2021-2022), 19,413 young people viewed the questionnaire, the participation rate was 92.1%, completion rate was 83.9%. The reasons for drop-out were primarily technical problems in the schools (e.g. instable internet connection, device-based time-out settings) and the lack of time to complete the questionnaire during the lessons.”

We hope we have addressed your comment properly.